# The prevalence of chronic pain in adolescents in Central Switzerland: A cross- sectional school-based study protocol

Helen Schwerdt[1,2]*, Guillaume Christe[1], Joshua W. Pate[3], Catherine Blake[2,4,5], Keith M. Smart[2,4,5]

1 Department of Physiotherapy, HESAV School of Health Sciences, HES-SO University of Applied Sciences and Arts Western Switzerland, Lausanne, Switzerland, 2 UCD School of Public Health, Physiotherapy and Sport Science, University College Dublin, Dublin, Ireland, 3 Graduate School of Health, University of Technology Sydney, Sydney, New South Wales, Australia, 4 Physiotherapy Department, St. Vincent's University Hospital, Dublin, Ireland, 5 UCD Centre for Translational Pain Research, University College Dublin, Dublin, Ireland

* helen.schwerdt@ucdconnect.ie

**Data Availability Statement:** Deidentified research data will be made publicly available when the study is completed, the adolescents accepted and published.

## Abstract

### Background

Chronic pain is associated with substantial personal suffering and societal costs and is a growing healthcare concern worldwide. While chronic pain has been extensively studied in adults, limited data exists on its prevalence and impact in adolescents. Understanding the prevalence and impact of chronic pain and pain beliefs in adolescents is crucial for developing effective prevention and treatment strategies. This study aims to estimate the prevalence, characteristics, and impact of chronic pain, and explore adolescents' knowledge and beliefs about pain.

### Methods

This is an observational cohort study of school-going adolescents aged 11 to 17 years in Central Switzerland. The study will estimate the point prevalence, characteristics (location, intensity, frequency, duration) and impact (PROMIS Pediatric Short Form v2.0 –Pain Interference Scale, PPIS) of chronic pain in school-going adolescents. We will also measure and investigate pupils' beliefs about pain (Concept of Pain Inventory (COPI)). Data will be collected through manual and digital self-report questionnaires and from participants in primary, secondary, and high schools between September 2023 and January 2024.

### Analyses

The primary analyses will utilise descriptive statistics to estimate the point prevalence, characteristics, and impact of chronic pain. Secondary analyses will analyse associations and correlations between chronic pain, impact of pain and beliefs about pain.

**Funding:** This study is funded by a Swiss – Irish Joint Doctoral Programme in Health Sciences (Health Sciences PhD) scheme.

**Competing interests:** I have read the journal's policy. HS, GC, and CB have no conflicts of interests to declare. KMS has received a conference fee waiver from the European Pain Federation (EFIC) (EFIC; Budapest, September 2023). JWP, also author of this manuscript has the following competing interests: he has received speaker fees for presentations on pain and rehabilitation. He receives royalties for books on pain education.

## Outcomes

This study will provide an estimate of the prevalence, characteristics and impact of chronic pain in adolescents in Central Switzerland and a measure of adolescents' understanding and beliefs about pain. In doing so, this study will provide insights into the scale of chronic pain as a public health concern. By understanding adolescents' pain beliefs and their influence on pain experience, this study can contribute to the development of educational approaches to enhance adolescents' knowledge and understanding of pain in order to optimise the prevention and treatment of chronic pain in adolescents. The findings may be useful to healthcare professionals and funders, policymakers, and researchers involved in the prevention, assessment, and treatment of pain in adolescents.

## Introduction

Chronic pain is responsible for up to six billion Swiss Francs in direct and indirect costs in Switzerland each year [1] and is a growing problem worldwide [2–4]. In recognition of the prevalence and impact of chronic pain, chronic pain has been included in the new International Classification of Diseases (ICD-11) [4].

Chronic pain is a serious public health concern among adolescents, leading to diminished quality of life, absenteeism from school and greater use of medication [5–9]. Importantly, the early onset of chronic pain in youth increases the risk of chronic pain in adulthood [10]. Global estimates of the prevalence of chronic pain in adolescents from aged 11 to 17 years old vary from 3 to 37% [5,9,11,12]. This variability may be explained by different definitions of chronic pain, and data collection methods [9,12]. High quality prevalence studies have been recommended in order to provide more accurate estimates of the prevalence of chronic pain in adolescents [9,12,13]. In Switzerland, there is a lack of data on the prevalence of chronic pain in adolescents [14]. The latest estimate suggested a nationwide Swiss prevalence of 3% of chronic pain in adolescents [15]. However, this indirect estimate was based on the perception of paediatricians rather than a sample of Swiss adolescents directly [15]. Therefore, an estimate of the prevalence, characteristics and impact of chronic pain based on a sample of Swiss adolescents is urgently required.

In recent decades, it has become increasingly evident that knowledge and beliefs about pain can contribute to the duration of pain and pain-related disability in adults [16–19] and in adolescents [20,21]. Examples of common misconceptions about pain in adolescents are that pain always signifies harm, that the body needs rest and avoidance to heal, and that emotions do not influence pain [20,21]. These conceptions are connected to expectations, which in turn shape the experience of pain [22–26]. In addition to increased pain, expectations can also lead to avoidance behaviour [27–30]. These beliefs are at odds with contemporary understanding of pain science. While the majority of pain belief research has focused on adults, limited studies have identified the existence of some inaccurate and unhelpful beliefs in adolescents [28,31–33]. Further data is required to expand our understanding of conceptions of pain in adolescents [34].

Investigating the prevalence and impact of chronic pain and pain beliefs in adolescents will provide much-needed data and insights into the extent and nature of chronic pain in adolescents as a public health concern. Investigating adolescents' knowledge and understanding of pain may help to develop educational strategies that contribute to the prevention and treatment of chronic pain in adolescents.

### Aims and objectives

The main aim of this project is to estimate the prevalence and impact of chronic pain in adolescents and to measure adolescents' knowledge and beliefs of pain. The primary objectives are to:

1. Estimate the point prevalence of chronic pain among adolescents between 11 and 17 years of age in Central Switzerland.

2. Investigate the characteristics (pain locations, duration, frequency, intensity) and impact of chronic pain in those adolescents experiencing it.

A secondary objective is to investigate adolescents' knowledge and beliefs about pain; and to explore associations and correlations between chronic pain, the impact of pain, beliefs about pain, age and gender.

## Methods

This study has been registered on the Open Science Framework [35]. This study is being undertaken in part fulfilment of a Doctor of Philosophy (PhD) degree at University College Dublin (UCD) in partnership with Haute Ecole de Santé Vaud (HESAV, HES-SO). As a principal investigator (PI) (HS) leads the study, with guidance and support from an academic supervisor (KS) and co-supervisor (GC). The study is also supported by an international collaborator with subject expertise (JP) and an academic with statistical expertise (CB). The study is overseen by a Research Study Panel with an independent chairperson.

### Study design

This observational cohort study will employ a cross-sectional design. Data collection started in September 2023 and will last for approximately five months.

### Participants

Eligible participants include students aged 11 to 17 years attending classes in the second and third education cycle in Central Switzerland. The exclusion criteria for this study includes adolescents who decline to consent, and adolescents who cannot comprehend verbal instructions and/or read study documents in the German language.

### Setting

Participants will be recruited from primary, secondary and high schools in the cantons of Lucerne, Zug, Schwyz and Aargau, that is from the German speaking region in Central Switzerland. There is no evidence for different pain prevalence compared to other German-speaking cantons in Switzerland. All participating schools will be mixed/single gender, publicly funded and non-fee paying, and located in a mix of urban and rural areas of Central Switzerland. Study participants with an age range between 11 and 17 years will be recruited from participating schools. Currently, four schools with about 400 possible adolescents have provisionally confirmed their participation.

### Recruitment

The participating schools were contacted via mail to invite them to take part in the study. The objectives of the study and the data collection procedure were presented to the school principals to provide them with a comprehensive understanding of the study. The school principals

have agreed to assist with recruiting participants. The study was promoted through mail advertisements within the participating schools. Once the teachers agreed to support the study, they received detailed information about the process. Schools and teachers have the option to withdraw their commitment to support the study at any time. In the event that a school decides to withdraw, possibly due to excessive effort required, other schools within the same region will be approached and invited to participate instead.

During official school class time, the teachers will verbally introduce the study to the pupils and provide them with the following documents:

- Adolescents aged 11–13 will receive an adapted written information document, the written information document for the parent/ legal guardian and an invitation for the pupils and the parent/legal guardian to attend study information sessions (3 documents).

- Adolescents aged 14–17 receive another adapted written information document, the written information document for the parent/ legal guardian and an invitation for the pupils and the parent/guardian to attend study information sessions (3 documents).

The parents (or legal guardian) of the pupils will be introduced to the study via a study information letter and two optional online information events. School principals, teachers, parents/legal guardians and pupils will be able to contact the PI (HS) who will discuss the study and ask any questions via mail, telephone or an information session. The information session will take place online and in the evening on two different days within at least one week of participant enrolment. At the information event, scheduled to last approximately 45 minutes, the PI will present the aims and objectives of the project and what participation will involve.

Parents/legal guardians and pupils will be informed that study participation is entirely voluntary and that their children are free, without justification, to decline to participate or withdraw from the study at any time. Before data collection commences, the PI will explain the procedure in plain simple language to the adolescents again and screen them for eligibility to ensure that participants meet the inclusion criteria. The screening procedure will involve verbally asking the child if they can read and write in the German language. All pupils receive the same online link. As an additional verification of willingness to participate, adolescents can decide to complete the survey or a reading substitute task. If a participant chooses the reading substitute task, a text previously chosen by the teacher will appear on the screen instead of the survey.

## Ethics

For the majority of participants, this study carries no inherent risks. We acknowledge that completing questionnaires about chronic pain could cause emotional distress for some individuals. Should this occur, the primary investigator (HS) and the school teachers will liaise with participant and their parents/legal guardians to ensure that the healthcare needs are appropriately assessed. Furthermore, participants will be informed that they are free to withdraw from the study at any time and that there are no direct benefits to their participation. The study has been approved by the ethics committee for the northwest and central part of Switzerland (Ethikkommission Nordwest- und Zentralschweiz, EKNZ) (BASEC number: 2023–00891).

## Data collection

The PI (HS) will be present in the classroom together with the teacher to administer the link to the questionnaires or reading substitute task to all adolescents. This also ensures that direct

enquiries can be made and compliance with the study protocol. Excluded children adolescents receive help from HS and the teacher to decide on the substitute task. Participating adolescents will be invited to complete questionnaires related to i) their demographics (age, gender, class grade), ii) the absence/presence of chronic pain, iii) pain characteristics and impact (those with chronic pain only), and iv) knowledge and beliefs about pain. The demographic data and the pain-related questionnaires will be completed via the cloud-based survey tool LimeSurvey. Piloting the questionnaires ensured that the questions were comprehensible and took approximately ten minutes to answer.

During the data collection period anonymised data will be stored digitally in the LimeSurvey cloud and will be accessible to the research team only. After each data collection session the data will be exported to and stored in a password protected folder on the Haute École de Santé Vaud in Switzerland for ten years. All data will be handled in accordance with current Swiss legislation [36]. On completion of the study we will make our anonymised data available without restriction on the Open Science Framework [35].

## Outcome measures

Sociodemographic data will be collected, including age (in whole years), gender (female, male, other, prefer not to say), and class grade (5th to 6th class primary school, 1st, 2nd and 3rd class upper school).

**Primary outcomes.** The IASP defines chronic pain as "pain that persists or recurs for longer than 3 months" [37]. For the purpose of this study, we operationally defined the presence of chronic pain as the self-reported experience of recurrent or persistent pain for three or more months and a frequency of at least weekly [38–42]. This definition was chosen because it is the most commonly used and can therefore be compared with other studies from the field [9,12].

The presence/absence of chronic pain will be determined by participants' self-reported responses to the following two questions:

- "Chronic pain is defined as pain that is recurrent or persistent for 3 months or longer. Do you feel chronic pain?" Response options: "No, I don't feel chronic pain", "Yes, I have felt pain that has lasted 3 months or longer."

- "How often do you feel this pain?" Response options: "daily"; "weekly"; "monthly", or "rarely" [38,40,41].

Those adolescents with chronic pain according to the operational definition will be invited to provide self-report additional information in order to characterise their pain, including:

- The location and total number of sites of their chronic pain, assessed via two body charts. The body chart is divided into 11 specific locations (head, neck, chest, shoulders, back, arms, hands, bottom/hips, belly/pelvis, legs, and feet) and an "other" category. For the first body chart, the child will be asked to mark the region in which they feel their most bothersome chronic pain. A second body chart will assess the number of chronic pain sites by the question: "Are there other areas where you have felt pain for 3 month or longer". The chronic pain sites have a score range from 0 to 12 based on axis I of the IASP Classification of Chronic Pain [43] and has been used in previous studies investigating pain in adolescents [5,40,44]. Its reliability and validity are unknown.

- Usual pain intensity by asking, "How intense the pain feels usually?", using an 11-point Numerical Rating Scale (NRS-11) (0 = no pain to 10 = the worst pain you can imagine". This question provides a generic pain intensity and does not distinguish between pain at rest or

activity [45]. The NRS-11 has been shown to provide reliable and valid scores when used with adolescents to assess current pain [46]. A high level of agreement in pain intensity ratings between the first and second measurements (Limits of Agreement: -0.9 to 1.2, 95% CI), indicating good reliability [46]. The NRS-11 has good convergent validity showing moderate to high correlations with other pain intensity scales [46]. Discriminant validity was supported by moderate correlations with measures of pain unpleasantness (r = 0.63) and functional impairment (r range: 0.35 to 0.43). The NRS-11 has demonstrated moderate correlations with measures of disability (r range: 0.22 to 0.39), psychological well-being (r range: 0.46 to 0.66), and pain interference (r = 0.62) [46]. Threshold values have been identified. Average scores of 3, 6, and 8 on the NRS-11 correspond to mild, moderate, and severe pain levels, respectively [46]. The NRS-11 has been used in a number of pain-related pediatric epidemiological studies [40–42].

- Pain duration by asking:" For how long have you felt this pain?", with the response options: "3–6 months", "6–12 months", "1–2 years" or "over 2 years". The validity and reliability of this retrospective question in adolescents is unknown. It has been used in numerous studies in adolescents [41,47–49] and will facilitate comparisons of results.

- The impact of their pain using the Patient Reported Outcome Measures Information System —Pediatric Pain Interference Scale (PROMIS-PPIS) short version in German [50]. PROMIS-PPI assesses self reported impairments in physical, psychological, and social functioning. The PROMIS-PPIS is an 8-item questionnaire studied in chronic pain children and adolescents, aged 2–17 years old. Scores range from 0 to 32 with higher scores indicating greater pain-related functional impairment. The PROMIS-PPIS is reliable [50] and has good internal consistency [51].

**Secondary outcome.**  *Pain knowledge and beliefs.* Adolescents' knowledge of and beliefs about pain will be measured in adolescents with and without chronic pain using the Concept of Pain Inventory in German (COPI- GER). The COPI- GER was initially designed for children and adolescents aged 8 to 12 years and assesses a child's concept of pain and therefore their beliefs and knowledge of pain [31]. Children indicate their level of agreement with 14 statements using a 5-point Likert scale ('Fully agree, 'agree', 'unsure', 'disagree' or 'Do not agree at all'). Scores range from 0 to 56 with higher scores indicating greater alignment with contemporary pain science knowledge. The COPI may identify potential misconceptions in a child's concept of pain and thereby facilitating the individualized targeting of pain science education [31]. In a content validation process conducted with 13 experts, the "Item-based content validity index" (I-CVI) ranged from 0.85 to 1.00, while the "Scale-based content validity index" (S-CVI) yielded an acceptable score of 0.93 [52]. The construct validity was confirmed by good goodness-of-fit indexes. Specifically, the Normed Fit Index, Incremental Fit Index, and Comparative Fit Index were all above 0.90, while the Root Mean Square Error of Approximation was less than 0.08 [32]. The total Cronbach's alpha was 0.78 [31,32], indicating good reliability. Validation of the German version is currently underway.

*Exploratory outcomes.* We will explore for any associations and correlations between the presence/absence of chronic pain, the impact of pain and adolescents' knowledge and beliefs about pain.

## Statistical analysis plan

Data will be entered, cleaned, and analysed using the Statistical Packages for the Social Sciences (SPSS), currently version 27. An estimate of the point prevalence of chronic pain will be

reported, with a 95% confidence interval. Adolescents' demographics, pain characteristics, PROMIS-PPIS and COPI-GER scores will be reported for the entire cohort and according to presence/absence of chronic pain, age and gender using standard descriptive statistics. Continuous data will be reported using mean scores, ranges, and standard deviations. Logistic regression models will be constructed to first establish the association between the presence of chronic pain, including PROMIS-PPIS and demographic factors (age, gender). Next the relationship between chronic pain and total COPI-GER score as well as individual items in the child's COPI-GER, independent of age and gender, will be explored with logistic regression models. Results will be expressed as odds ratios.

Linear regression models will be used to explore the association between the pain interference (PROMIS-PPIS) and the pain knowledge and beliefs (COPI-GER) in adolescents with and without chronic pain, with further analysis by age and gender. Sensitivity analyses will be performed to assess the robustness of the results, such as excluding participants or outliers.

**Missing data.** Missing data will be handled according to the framework described by Lee et al. [53]. The number/proportion of missing values for each variable will be reported, along with any assumptions made regarding the cause of missingness. Patterns of missing data and likely reasons will be explored. The validity of a complete records analysis will be considered. If a complete records analysis is deemed invalid and the assumption is made that data are 'missing at random', multiple imputation will be considered to reduce bias and improve precision. For instance, if data for a given outcome measure are missing from ≥10% of participants, missing data will be imputed using the multiple imputation by chained equations method [54]. The number of imputations will be determined based on the percentages of missing values, and the results for the imputations will be pooled using Rubin's rule.

**Sample size.** Previous studies have estimated that 3–38% of adolescents suffer from chronic pain. Assuming that 25% of our sample have chronic pain, this suggests that for a total of 288 responses, 95% confidence limits will be 20% and 30% with a precision of 5% (Sample size for prevalence surveys, © 2003–2005 Philippe Glaziou). With an estimated response rate of 80%, it is aimed to include 360 adolescents. The calculation is based on a population size of 86424 adolescents between 11 and 17 years in central part of Switzerland (canton Luzern, Aargau, Schwyz and Zug) in 2021.

## Timeline

Preparations for data collection and information events took place in August and September 2023. Data collection commenced in September 2023 and is expected to be completed in January 2024. Data analysis will begin on completion of the data collection.

## Discussion

This study will attempt to provide a more precise estimate of the prevalence, characteristics and impact of chronic pain in adolescents aged between 11 and 17 years in Central Switzerland compared to previous research [15]. Specifically, this study will provide estimates of the severity, duration, and location of chronic pain in adolescents and the extent to which it impacts on their physical, psychological and social functioning.

Furthermore, the study will assess the adolescents' knowledge and beliefs about pain. Unhelpful beliefs and misconceptions about pain may contribute to maladaptive behaviours and the development of chronic pain. Identifying specific misconceptions about pain could inform future studies that design interventions to target these misconceptions and better align adolescents' understanding of pain with contemporary pain science. Such understanding may ultimately contribute to the prevention and treatment of chronic pain in adolescents. Future

projects can utilize the findings to design interventions that target and rectify these misconceptions.

### Limitations

This study has several potential limitations. One limitation is the non-randomized recruitment process. It should be noted that recruiting schools to participate in research studies can be challenging. However, the study aims to ensure a sufficient sample size to make valid inferences about the population.

Even though the observational study design is the most appropriate design to answer the research questions, this design is prone to several internal validity problems, such as self-report bias, selection bias, information bias, and reporting bias [55].

The potential limitations of self-reporting when using PROMs to measure our outcomes of interest are acknowledged [56]. Specifically, the pupils are asked how long they have been in pain, which could lead to a recall bias. To reduce this bias, a reference point will be discussed with the teacher beforehand to help the adolescents accurately estimate the duration of the past three months. Furthermore, age- appropriate validated questionnaires will be utilized.

The study outcomes are based, in part, on existing pain-related instruments used in similar studies (e.g. pain intensity, pain locations). The reliability and validity of our exact instrument has not been tested but is inferred from the available psychometric data upon which the original questionnaires are based.

The scope of our study does not enable us to collect data on medication usage, the influence of any perceived school-related pressures, the prevalence of chronic pain among parents or socio-economic status. The extent to which these variables might confound our findings is not known.

The selection and recruitment procedure could lead to a selection bias, towards those with more severe or chronic pain. To reduce this bias, the project will use a standard recruitment process to ensure a representative sample of the population. Information bias will be limited by using the most valid and reliable outcome measures available. In order to minimise reporting bias we have published this study protocol and will report any and all deviations from the protocol. Finally, our study will recruit participants from schools in Central Switzerland and as such our findings may not generalise to other geographical locations.

### Dissemination

The study will be reported according to 'The Strengthening the Reporting of Observational Studies in Epidemiology (STROBE)' guidelines [57]. We will aim to publish our findings in pain-related, peer-reviewed journals, within 12 months of completing the study. Findings will be further disseminated via relevant clinical and academic conferences (held by the Swiss Pain Society and the European Pain Federation for example), pain-related public and patient advocacy groups, and social media. We will present our findings to the adolescents, parents, teachers and school administrators at a post-study online evening seminar. We aim to make our data FAIR, i.e., findable, accessible, interoperable, and reusable [58].

### Supporting information

**S1 EthicsGer. The ethical approval for this study is available in German; please refer to document S1 EthicsGer provided as supporting information.**
(PDF)

**S2 EthicsGer. For better comprehension, a translation into English has been prepared based on the original ethical approval; please see document S2 EthicsEng also provided as supporting information.**
(DOCX)

## Acknowledgments

The authors intend to involve patients and public insight partners from the Lucerne, Zug, or Aargau region and will provide a detailed description of their contribution and influence.

## Author Contributions

**Conceptualization:** Helen Schwerdt, Guillaume Christe, Keith M. Smart.

**Methodology:** Helen Schwerdt, Guillaume Christe, Joshua W. Pate, Catherine Blake, Keith M. Smart.

**Project administration:** Helen Schwerdt.

**Resources:** Helen Schwerdt.

**Supervision:** Guillaume Christe, Keith M. Smart.

**Validation:** Keith M. Smart.

**Writing – original draft:** Helen Schwerdt.

**Writing – review & editing:** Guillaume Christe, Joshua W. Pate, Catherine Blake, Keith M. Smart.

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
