## [Decision Letter · Decision Letter 0]

20 Oct 2023

PONE-D-23-25399The Prevalence of Chronic Pain in Children and Adolescents in Central Switzerland: A Cross- Sectional School-based Study Protocol.PLOS ONE

Dear Dr. Schwerdt,

Thank you for submitting your manuscript to PLOS ONE. After careful consideration, we feel that it has merit but does not fully meet PLOS ONE’s publication criteria as it currently stands. Therefore, we invite you to submit a revised version of the manuscript that addresses the points raised during the review process.

We look forward to receiving your revised manuscript.

Kind regards,

Renato S. Melo, PhD

Academic Editor

PLOS ONE

Journal Requirements:

3. Ethics statement only appears at the end of the manuscript:

Your ethics statement should only appear in the Methods section of your manuscript. If your ethics statement is written in any section besides the Methods, please move it to the Methods section and delete it from any other section. Please ensure that your ethics statement is included in your manuscript, as the ethics statement entered into the online submission form will not be published alongside your manuscript.

Reviewers' comments:

Reviewer's Responses to Questions

**Comments to the Author**

1. Does the manuscript provide a valid rationale for the proposed study, with clearly identified and justified research questions?

Reviewer #1: Partly

Reviewer #2: Yes

2. Is the protocol technically sound and planned in a manner that will lead to a meaningful outcome and allow testing the stated hypotheses?

Reviewer #1: Yes

Reviewer #2: Yes

3. Is the methodology feasible and described in sufficient detail to allow the work to be replicable?

Reviewer #1: No

Reviewer #2: Yes

4. Have the authors described where all data underlying the findings will be made available when the study is complete?

Reviewer #1: Yes

Reviewer #2: Yes

5. Is the manuscript presented in an intelligible fashion and written in standard English?

Reviewer #1: Yes

Reviewer #2: Yes

6. Review Comments to the Author

You may also provide optional suggestions and comments to authors that they might find helpful in planning their study.

Reviewer #1: Dear authors,

One of the primary aims of drafting a research protocol is to provide a comprehensive description of the planned study, encompassing its objectives, design, methodology, statistical considerations, and organisational aspects pertaining to a clinical study. Additionally, the protocol serves the purpose of safeguarding the well-being of trial participants and preserving the integrity of the data obtained. Given that these problems have not been adequately addressed, I request the authors to revise their work in accordance with the constructive criticism offered.

The authors assert in the introductory part that chronic pain is a prevalent manifestation of musculoskeletal disorders (MSD). Following this assertion, there appears to be no further instances in which the authors ascribe ongoing pain to MSD. It is unclear whether the authors' intention is to assess the prevalence of chronic pain specifically attributed to MSD or chronic pain in general. I am also interested in ascertaining the precise details of the information that was communicated to the school administration, as well as the individual responsible for collecting the data . I comprehend that the authors intend to employ a body chart as a means of discerning the specific locations of pain originating from various regions of the human body. However, it remains unclear if the authors intend to gather data on pain severity during periods of rest or activity. Additionally, the authors state that children will be selected for recruitment based on certain inclusion criteria, although these criteria are not explicitly outlined.

The protocol has weaknesses in the following aspects: The authors has not supplied any information regarding the development of the questionnaire, its suitability for young people and adolescents, the psychometric evaluation of the questionnaire, or the total time required to complete it. It is imperative to provide information regarding the duties of any committees involved in the study, as well as specifying the individuals responsible for overseeing the study. Is there a limitation on the concurrent healthcare services offered to the participants during the data gathering process? In light of the findings of the study, what methods have been formulated for post-care? What are the proposed strategies for disseminating the findings or results of this study?

In addition, there are some minor details that should be taken into consideration.

• Sentences 62-63: Does this data have global applicability or is it specific to Switzerland?

• Please provide a more detailed explanation of the erroneous and unhelpful assumptions around pain experienced by children.

• Sentence 64: What is the significance of regional differences?

• Sentences 84-87: Kindly review again.

• Given the dearth of prevalence estimates, what are the reasons for confining the investigation to Central Switzerland?

• Who will collect the data, how the data will be managed securely, and who will feed the data into the computers?

• Given that the study adopts a cross-sectional design, the authors posit the elements that they perceive to be responsible for the occurrence of missing data. What measures would be implemented to ensure the participants' compliance with the study protocol?

• Please add a section that will discuss the methodology for data collection and management, with a particular focus on safeguarding personal information and ensuring confidentiality.

• Kindly present a section elucidating the authors' strategies to mitigate the impact of confounding variables, encompassing the utilisation of current medications.

• Sentence 300 need rephrasing.

Regards.

Reviewer #2: Thank you to the authors for their submission. The paper addresses a very important clinical and research topic and has relevance in the field. The study limitations were well presented and justified. As a suggestion, authors could also add socioeconomic level data to somehow consider the recruitment bias.

7. PLOS authors have the option to publish the peer review history of their article (what does this mean?). If published, this will include your full peer review and any attached files.

Reviewer #1: **Yes: **G Shankar Ganesh

Reviewer #2: No

---

## [Author Response · Author response to Decision Letter 0]

29 Nov 2023

Dear Reviewers and Editor,

Re: Manuscript PONE-D-23-25329

The Prevalence of Chronic Pain in Children and Adolescents in Central Switzerland: A Cross- Sectional School-based Study Protocol.

Thank you to the reviewers for their time and attention in reviewing this manuscript. Our responses to the reviewers’ comments are detailed below. We hope they find these acceptable. New or amended text is indicated in red coloured text (see response to reviewers document), as in the revised manuscript. No red colour is possible in the following text. You will only see the text in the corrected version. 

Reviewer #1 

1) The authors assert in the introductory part that chronic pain is a prevalent manifestation of musculoskeletal disorders (MSD). Following this assertion, there appears to be no further instances in which the authors ascribe ongoing pain to MSD. It is unclear whether the authors' intention is to assess the prevalence of chronic pain specifically attributed to MSD or chronic pain in general. 

We mean to estimate the prevalence and impact of chronic pain in adolescents in the broadest sense with respect to aetiology. We have removed all references to ‘musculoskeletal pain’ and now use the general term ‘chronic pain’ throughout. 

We changed the following text in the abstract: Chronic pain is associated with substantial personal suffering and societal costs and is a growing healthcare concern worldwide. [Abstract. Lines 26/27]

2) I am also interested in ascertaining the precise details of the information that was communicated to the school administration, as well as the individual responsible for collecting the data. 

We have included the following to clarify the information that was conveyed to the school administrators “The objectives of the study and the data collection procedure were presented to the school principles to provide them with a comprehensive understanding of the study.” [Recruitment. Lines 179/180]

3) I comprehend that the authors intend to employ a body chart as a means of discerning the specific locations of pain originating from various regions of the human body. However, it remains unclear if the authors intend to gather data on pain severity during periods of rest or activity. 

We intend to gather a generic measure of pain severity as ‘usually’ perceived and experienced and does not distinguish between rest or activity. To clarify this we have added the following text: “Usual pain intensity by asking, “How intense the pain feels usually?”, using an 11-point Numerical Rating Scale (NRS-11) (0 = no pain to 10 = the worst pain you can imagine”. This question provides a generic pain intensity and does not distinguish between pain at rest or activity (46)”. [Primary outcomes. Lines 306-314]

4) Additionally, the authors state that children will be selected for recruitment based on certain inclusion criteria, although these criteria are not explicitly outlined.

The eligibility criteria are reported in the section Participants, lines 158-162, as: “Eligible participants include students aged 11 to 17 years attending classes in the second and third education cycle in Central Switzerland. The exclusion criteria for this study includes adolescents who decline to consent, and adolescents who cannot comprehend verbal instructions and/or read study documents in the German language.” We think the eligibility (i.e. inclusion) criteria have been adequately reported; no amendments made. [Participants. Lines 158-162]

5) The protocol has weaknesses in the following aspects: The authors has not supplied any information regarding the development of the questionnaire, its suitability for young people and adolescents, the psychometric evaluation of the questionnaire, or the total time required to complete it. 

In the sections ‘Primary Outcomes’ and ‘Secondary outcomes’ we report that the questionnaire instrument was developed from and based on pre-designed questionnaires from a number of cited studies from within the field. We also report their psychometric properties where data is available; and we acknowledge where it is not. We have added the following text to the ‘Limitations’ section: “The study outcomes are based, in part, on existing pain-related instruments used in similar studies (e.g., pain intensity, pain locations). The reliability and validity of our exact instrument has not been tested but is inferred from the available psychometric data upon which the original questionnaires are based.” We hope this is acceptable. [Limitations. Lines 467- 473]

To clarify the time to complete the questionnaire we included the following information in the section Outcome measures: “Piloting the questionnaires ensured that the questions were comprehensible and took approximately ten minutes to answer.” [Outcome measures. Lines 268/269]

6) It is imperative to provide information regarding the duties of any committees involved in the study, as well as specifying the individuals responsible for overseeing the study. 

More information on the committees involved are given in the section Methods, lines 148-154. The paragraph has been amended to read as follows: “This study is being undertaken in part fulfilment of a Doctor of Philosophy (PhD) degree at University College Dublin (UCD) in partnership with Haute Ecole de Santé Vaud (HESAV, HES-SO). As a principal investigator (HS) leads the study, with guidance and support from an academic supervisor (KS) and a co-supervisor (GC). The study is also supported by an international collaborator with expertise (JP) and an academic with statistical expertise (CB). The study is overseen by a research study panel with an independent chairperson.” [Methods. Lines 148-154]

7) Is there a limitation on the concurrent healthcare services offered to the participants during the data gathering process? In light of the findings of the study, what methods have been formulated for post-care? 

This is an epidemiology study in the Swiss school system. It is taken place independent of any health care services. There is no health care provided as part of the study.

8) What are the proposed strategies for disseminating the findings or results of this study?

Our strategies for disseminating the findings from this study have been described [Funding and Dissemination. Lines 489-494]. We have added the following text: “Findings will be further disseminated via relevant clinical and academic conferences (held by the Swiss Pain Society and the European Pain Federation for example), pain-related public and patient advocacy groups, and social media. We will present our findings to the children, parents, teachers and school administrators at a post-study online evening seminar.”

9) In addition, there are some minor details that should be taken into consideration. Sentences 62-63: Does this data have global applicability or is it specific to Switzerland?

The text has been amended to “Global estimates of the prevalence of chronic pain in school children from 11 to 17 years old varies from 3 to 37% (5,9,11,12).” [Introduction. Lines 70-72]

10) Please provide a more detailed explanation of the erroneous and unhelpful assumptions around pain experienced by children. 

In the introduction, the following sentence is added to clarify the erroneous and unhelpful assumptions around pain: “In recent decades, it has become increasingly evident that knowledge and beliefs about pain can contribute to the duration of pain and pain-related disability in adults (16–19) and in adolescents (20-21). ”Examples of common misconceptions about pain in adolescents are that pain always signifies harm, that the body needs rest and avoidance to heal, and that emotions doesn’t influence pain (20,21).” [Introduction. Lines 81-114]

11) Sentence 64: What is the significance of regional differences?

We have amended the text to: “This variability may be explained by different definitions of chronic pain, and data collection methods differences (9,12).” [Introduction. Lines 72/73]

13) Sentences 84-87: Kindly review again. 

We are unsure of the reviewer’s issue here and on reviewing the text we do not think any changes are required. However, we have swapped ‘children’ for ‘adolescents’.

14) Given the dearth of prevalence estimates, what are the reasons for confining the investigation to Central Switzerland?

Limited financial resources restricted the scope of the study, making it more feasible to focus on a specific region rather than attempting a broader nationwide investigation. Due to resource limitations, it was more practical to conduct the study within a local area, Central Switzerland, where logistical and operational challenges could be managed more effectively.

15) Who will collect the data, how the data will be managed securely, and who will feed the data into the computers? 

In the section Data collection the following paragraph clarifies the procedure: “One researcher (HS) will be present in the classroom together with the teacher to will administer the link to the questionnaires or reading substitute task to all adolescents. This also ensures direct enquiries can be made and compliance with the study protocol. Excluded adolescents receive help from HS and the teacher to decide on the substitute task. Participating adolescents will be invited to complete questionnaires related to i) their demographics (age, gender, class grade), ii) the absence/presence of chronic pain, iii) pain characteristics and impact (those with chronic pain only), and iv) knowledge and beliefs about pain. The demographic data and the pain-related questionnaires will be completed via the cloud-based survey tool LimeSurvey. 

During the data collection period anonymised data will be stored digitally in the LimeSurvey cloud and will be accessible to the research team only. After each data collection session the data will be exported and stored in a password protected folder on the server at Haute École de Santé Vaud in Switzerland for ten years. All data will be handled in accordance with current Swiss legislation (37). [Data collection. Lines 250-264]

16) Given that the study adopts a cross-sectional design, the authors posit the elements that they perceive to be responsible for the occurrence of missing data. What measures would be implemented to ensure the participants' compliance with the study protocol?

It has been clarified that the teacher and the PI are physically present in the room with the participants to ensure compliance with the study protocol. [Data collection. Lines 252/253]

17) Please add a section that will discuss the methodology for data collection and management, with a particular focus on safeguarding personal information and ensuring confidentiality. 

The section Data collection is specified as outline above. [Data collection. Lines 250-264]

18) Kindly present a section elucidating the authors' strategies to mitigate the impact of confounding variables, encompassing the utilisation of current medications.

It is not feasible to us to collect data on medication usage. We have added the text to the limitations in the section Limitations: “The scope of our study does not enable us to collect data on medication usage, the prevalence of chronic pain among parents or socio-economic status. The extent to which these variables might confound our findings is not known.” [Limitations. Lines 472-474]

19) Sentence 300 need rephrasing.

“Specifically, this study will provide estimates of the severity, duration, and location of chronic pain in adolescents and the extent to which it impacts on their physical, psychological and social functioning.” [Limitations. Lines 432-434]

Reviewer #2

1) (..) As a suggestion, authors could also add socioeconomic level data to somehow consider the recruitment bias.

Thank you for the suggestion. Given that children may not be able to accurately report this information, and considering the potential burden it would impose on parents, the decision was made not to attempt to collect socioeconomic data. We have added the text to the limitations in the section Limitations: “The scope of our study does not enable us to collect data on medication usage, the prevalence of chronic pain among parents or socio-economic status. The extent to which these variables might confound our findings is not known.” [Limitations. Lines 472-474]

General requirements: 

The name of the document is adapted to the guideline. 

“On completion of the study we will make our anonymised data available on the Open Science Framework.” [Methods. Lines 168] DOI: 10.17605/OSF.IO/Z3QF6. 

3. Ethics statement only appears at the end of the manuscript:

Your ethics statement should only appear in the Methods section of your manuscript. If your ethics statement is written in any section besides the Methods, please move it to the Methods section and delete it from any other section. Please ensure that your ethics statement is included in your manuscript, as the ethics statement entered into the online submission form will not be published alongside your manuscript.

The ethic statement is replaced to the methods [Line 223-232] and deleted in the Ethic, Funding and Dissemination Section. This section is renamed to Funding and Dissemination. [Funding and Dissemination. Lines 485]

Additional changes: 

1) According to the WHO's definition of age groups, pupils aged 11-17 years are considered adolescents. Therefore, the term 'children' has been revised to 'pupils and adolescents' throughout the manuscript. The title has also been revised: “The Prevalence of Chronic Pain in Adolescents in Central Switzerland: A Cross- Sectional School-based Study Protocol.“ [Title. Lines 1-3]

2) To clarify the aims in the abstract we changed the following text: “This study aims to estimate the prevalence, characteristics, and impact of chronic pain, and explore adolescents' knowledge and beliefs about pain.” [Abstract. Lines 30-32]

3) We have edited the following text for clarification: 

• “High quality prevalence studies have been recommended in order to provide more accurate estimates of the prevalence of chronic pain in adolescents (9,12,13). [Introduction. Line 62-64] 

• “The latest estimate suggested a nationwide Swiss prevalence of 3% of chronic pain in adolescents. [Introduction. Lines 78/79]

4) The description of the PROMIS PPI Questionnaire has been adapted for clarification. “The impact of their pain using the Patient Reported Outcome Measures Information System - Pediatric Pain Interference Scale (PROMIS-PPIS) short version in German (51). PROMIS-PPI assesses self-reported impairments in physical, psychological, and social functioning.” [Primary outcomes. Lines 334/348]

5) We asked twice about chronic pain locations. To clarify the specific location referred to in the first question, we rephrased the following sentence in the primary outcome section.: 

For the first body chart, the child will be asked to mark the region in which they feel their most bothersome chronic pain. [Primary outcomes. Line 300]

6) To clarify the funding scheme we added the following text: “This study is funded by a Swiss – Irish Joint Doctoral Programme in Health Sciences (Health Sciences PhD) scheme.” [Funding and Dissemination. Lines 515/516]

Thank you for re-reviewing our protocol for publication in PLOS ONE. 

Yours sincerely, 

Helen Schwerdt

---

## [Decision Letter · Decision Letter 1]

4 Dec 2023

PONE-D-23-25399R1The Prevalence of Chronic Pain in Adolescents in Central Switzerland: A Cross- Sectional School-based Study Protocol.PLOS ONE

Dear Dr. Schwerdt,

Thank you for submitting your manuscript to PLOS ONE. After careful consideration, we feel that it has merit but does not fully meet PLOS ONE’s publication criteria as it currently stands. Therefore, we invite you to submit a revised version of the manuscript that addresses the points raised during the review process.

We look forward to receiving your revised manuscript.

Kind regards,

Renato S. Melo, PhD

Academic Editor

PLOS ONE

Journal Requirements:

Reviewers' comments:

Reviewer's Responses to Questions

**Comments to the Author**

1. Does the manuscript provide a valid rationale for the proposed study, with clearly identified and justified research questions?

Reviewer #1: Yes

2. Is the protocol technically sound and planned in a manner that will lead to a meaningful outcome and allow testing the stated hypotheses?

Reviewer #1: Yes

3. Is the methodology feasible and described in sufficient detail to allow the work to be replicable?

Reviewer #1: Yes

4. Have the authors described where all data underlying the findings will be made available when the study is complete?

Reviewer #1: No

5. Is the manuscript presented in an intelligible fashion and written in standard English?

Reviewer #1: Yes

6. Review Comments to the Author

You may also provide optional suggestions and comments to authors that they might find helpful in planning their study.

Reviewer #1: Dear authors,

The revised manuscript has successfully addressed the majority of the issues that were raised. Nevertheless, it is necessary to revise the work for the English language. Please consider the following concerns during the process of revision:

• Correct spelling errors in sentences 61, 125, and so on.

• As an individual residing outside of the European Union, my knowledge of the enrolment status of all adolescents in educational institutions is limited. The primary objective of the authors is to assess the prevalence of pain among adolescents and not necessarily school going adolescents. The data for this study is exclusively collected from participants who attend school. The extent to which school-related pressures contribute to the perception of pain is uncertain. Either present it as a limitation or adjust the subject matter accordingly.

• Sentence 106–107: Given that the timing of this revision is far beyond September 2023, it is advisable to revise the tense where appropriate.

Regards.

7. PLOS authors have the option to publish the peer review history of their article (what does this mean?). If published, this will include your full peer review and any attached files.

Reviewer #1: No

---

## [Author Response · Author response to Decision Letter 1]

18 Dec 2023

18th December 2023

Dear Reviewers and Editor,

Re: Manuscript PONE-D-23-25399R1

The Prevalence of Chronic Pain in Adolescents in Central Switzerland: A Cross-Sectional School-based Study Protocol.

Thank you to the reviewers for their time and attention in reviewing this manuscript again. Our responses to the reviewers’ comments are detailed below. We hope they find these acceptable. New or amended text is indicated in red coloured text, as in the revised manuscript.

4. Have the authors described where all data underlying the findings will be made available when the study is complete?

The PLOS Data policy requires authors to make all data underlying the findings described in their manuscript fully available without restriction, with rare exception, at the time of publication. The data should be provided as part of the manuscript or its supporting information or deposited to a public repository. For example, in addition to summary statistics, the data points behind means, medians and variance measures should be available. If there are restrictions on publicly sharing data—e.g. participant privacy or use of data from a third party—those must be specified.

Reviewer #1: No

As a protocol, there are no data associated with this article. On completion of the study we will make our anonymised data available without restriction on the Open Science Framework. We report this as: “On completion of the study we will make our anonymised data available on the Open Science Framework (35).” [Data collection. Lines 188 - 190] 

Reviewer #1: Dear authors,

The revised manuscript has successfully addressed the majority of the issues that were raised. Nevertheless, it is necessary to revise the work for the English language. Please consider the following concerns during the process of revision:

• Correct spelling errors in sentences 61, 125, and so on.

• As an individual residing outside of the European Union, my knowledge of the enrolment status of all adolescents in educational institutions is limited. The primary objective of the authors is to assess the prevalence of pain among adolescents and not necessarily school going adolescents. The data for this study is exclusively collected from participants who attend school. The extent to which school-related pressures contribute to the perception of pain is uncertain. Either present it as a limitation or adjust the subject matter accordingly.

• Sentence 106–107: Given that the timing of this revision is far beyond September 2023, it is advisable to revise the tense where appropriate.

1. The manuscript has been corrected for English language as follows: 

Sentence 61: Global estimates of the prevalence of chronic pain in adolescents from aged 11 to 17 years old vary from 3 to 37% (5,9,11,12). [Introduction. Lines 60-62]

Sentence 125: The objectives of the study and the data collection procedure were presented to the school principals to provide them with a comprehensive understanding of the study. [Recruitment. Lines 125 - 127]

We have corrected the following paragraphs: 

“Examples of common misconceptions about pain in adolescents are that pain always signifies harm, that the body needs rest and avoidance to heal, and that emotions do not influence pain (20,21). These conceptions are connected to expectations, which in turn shape the experience of pain (22–26). In addition to increased pain, expectations can also lead to avoidance behaviour (27–30). These beliefs are at odds with contemporary understanding of pain science. While the majority of pain belief research has focused on adults, limited studies have identified the existence of some inaccurate and unhelpful beliefs in adolescents (28,31–33). Further data is required to expand our understanding of conceptions of chronic pain in adolescents (34).” [Introduction. Lines 73 - 81]

“Currently, 4 schools with about 400 possible adolescents have provisionally confirmed their participation.” [Setting. Lines 122 -123]

"School principals, teachers, parents/legal guardians and pupils will be able to contact the PI (HS) who will discuss the study and ask any questions via mail, telephone or an information session." [Recruitment. Lines 145 - 147]

“All pupils receive the same online link. As an additional verification of willingness to participate, adolescents can decide to complete the survey or a reading substitute task. If a participant will chooses the reading substitute task, a text previously chosen by the teacher will appear on the screen instead of the survey.” [Recruitment. Lines 158 - 161]

“The PI (HS) will be present in the classroom together with the teacher to administer the link to the questionnaires or reading substitute task to all adolescents.” [Data collection. Lines 173 - 174]

The following sentence has been moved from Outcome measures to the section Data collection: “Piloting the questionnaires ensured that the questions were comprehensible and took approximately ten minutes to answer.” [Data collection. Lines 181 - 182]

“For the purpose of this study, we operationally defined the presence of chronic pain as the self-reported experience of recurrent or persistent pain for three or more months and a frequency of at least weekly [38–42]. This definition was chosen because it is the most commonly used and can therefore be compared with other studies from the field [9,12].” [Primary outcomes. Lines 199 - 203]

“The presence/absence of chronic pain will be determined by participants’ self-reported responses to the following two questions:” [Primary outcomes. Lines 204 - 205]

“The chronic pain sites have a score range from 0 to 12 based on axis I of the IASP Classification of Chronic Pain [43] and has been used in previous studies investigating pain in adolescents [5,40,44]. Its reliability and validity are unknown.” [Primary outcomes. Lines 219 - 222]

“The NRS-11 has good convergent validity, as it showing moderate to high correlations with other pain intensity scales [46]” [Primary outcomes. Lines 230 - 231]

“Threshold values have been identified. Average scores of 3, 6, and 8 on the NRS-11 correspond to mild, moderate, and severe pain levels, respectively [46]. The NRS-11 has been used in a number of pain-related pediatric epidemiological studies to ask for the chronic pain intensity [40–42].” [Primary outcomes. Lines 236 - 239]

“It has been used in numerous studies in adolescents [41,47–49] and will facilitate comparisons of results.” [Primary outcomes. Lines 243 - 244]

“The construct validity was confirmed by good goodness-of-fit indexes.” [Primary outcomes. Lines 266 - 267]

“Next the relationship between chronic pain and total concept of pain (COPI-GER score) as well as individual items in the child’s COPI-GER, independent of age and gender, will be explored with logistic regression models. Results will be expressed as odds ratios.” [Statistical Analysis Plan. Lines 284 - 287]

“The calculation is based on a population size of 86424 adolescents between 11 and 17 years in central part of Switzerland (canton Luzern, Aargau, Schwyz and Zug) in 2021.” [Sample Size. Lines 310 - 312]

This study will attempt to provide a more precise estimate of the prevalence, characteristics, and impact of chronic pain in adolescents aged between 11 and 17 years in Central Switzerland compared to previous research [15]. Unhelpful beliefs and misconceptions about pain may contribute to maladaptive behaviours and the development of chronic pain. Identifying specific misconceptions about pain could inform future studies that design interventions to target and rectify these misconceptions and better align adolescents’ understanding of pain with contemporary pain science. Such understanding may ultimately contribute to the prevention and treatment of chronic pain in adolescents. Future projects can utilize the findings to design interventions that target and rectify these misconceptions. [Discussion. Lines 318 - 330]

“The potential limitations of self-reporting when using PROMs to measure our outcomes of interest are acknowledged [60.]” [Limitations. Lines 339 - 340]

“To reduce this bias, a reference point will be discussed with the teacher beforehand to help the adolescents accurately estimate the duration of the past three months.” [Limitations. Lines 341 - 343]

2. The study was declared as a school-based study in the title and in the methods. In order to specify the influence of school pressure on the results, which was not surveyed, the text has been adjusted in the ‘Limitations’ section as follows: “The scope of our study does not enable us to collect data on medication usage, the influence of any perceived school-related pressures, the prevalence of chronic pain among parents or socio-economic status. The extent to which these variables might confound our findings is not known.” [Limitations. Lines 349 - 352] 

We also have adapted the following text in the abstract: “This is an observational cohort study of school-going adolescents aged 11 to 17 years in Central Switzerland. The study will estimate the point prevalence, characteristics (location, intensity, frequency, duration) and impact (PROMIS Pediatric Short Form v2.0 – Pain Interference Scale, PPIS) of chronic pain in school-going adolescents.” [Abstract. Lines 32 - 36]

3. Sentence 108-109 is in the manuscript. As the data collection is currently ongoing, we wrote the manuscript in future tense. 

We have adapted the following text in line with progress:

 “Data collection started in September 2023 and will last for approximately five months.” [Study design. Lines 108 - 109]

“Preparations for data collection and information events took place in August and early September 2023. Data collection commenced in September 2023 and is expected to be completed in January 2024. Data analysis will begin on completion of the data collection.” [Timeline. Lines 314 - 316]

Thank you for re-reviewing our protocol for publication in PLOS ONE. 

Yours sincerely, 

Helen Schwerdt, PhD cand.

Mail: Helen.schwerdt@ucdconnect.ie

Tel. : +41 78 208 15 33

---

## [Decision Letter · Decision Letter 2]

27 Dec 2023

The Prevalence of Chronic Pain in Adolescents in Central Switzerland: A Cross-Sectional School-based Study Protocol.

PONE-D-23-25399R2

Dear Dr. Schwerdt,

We’re pleased to inform you that your manuscript has been judged scientifically suitable for publication and will be formally accepted for publication once it meets all outstanding technical requirements.

Kind regards,

Renato S. Melo, PhD

Academic Editor

PLOS ONE

Additional Editor Comments (optional):

Reviewers' comments:

Reviewer's Responses to Questions

**Comments to the Author**

1. Does the manuscript provide a valid rationale for the proposed study, with clearly identified and justified research questions?

Reviewer #1: Yes

2. Is the protocol technically sound and planned in a manner that will lead to a meaningful outcome and allow testing the stated hypotheses?

Reviewer #1: Yes

3. Is the methodology feasible and described in sufficient detail to allow the work to be replicable?

Reviewer #1: Yes

4. Have the authors described where all data underlying the findings will be made available when the study is complete?

Reviewer #1: Yes

5. Is the manuscript presented in an intelligible fashion and written in standard English?

Reviewer #1: Yes

6. Review Comments to the Author

You may also provide optional suggestions and comments to authors that they might find helpful in planning their study.

Reviewer #1: Dear author,

There are still typographical errors in the work.

A few of the errors to be modified include:

• Sentence 58: In youth, chronic pain is a serious public health problem leading to worse quality of life, more days absent from school, and more medication intake.

• Sentence 60: 'Global estimates of the prevalence of chronic pain in adolescents aged 11 to 17 vary from 3 to 37% [5, 9, 11, 12]'.

• Sentence 62: This variability may be explained by different definitions of chronic pain and data collection methods.

• Rephrase sentence 102 as 'A principal investigator (PI) (HS) leads the study with guidance and support from an academic supervisor (KS) and co-supervisor (GC)'.

Regards.

7. PLOS authors have the option to publish the peer review history of their article (what does this mean?). If published, this will include your full peer review and any attached files.

Reviewer #1: No

---

## [Editor Report · Acceptance letter]

31 Jan 2024

PONE-D-23-25399R2 

PLOS ONE

Dear Dr. Schwerdt, 

I'm pleased to inform you that your manuscript has been deemed suitable for publication in PLOS ONE. Congratulations! Your manuscript is now being handed over to our production team.

Kind regards, 

on behalf of

Dr. Renato S. Melo 

Academic Editor

PLOS ONE